# A Methodology for Generating Service Areas That Accounts for Linear Barriers

**Paweł Flisek** [†] [ID] **and Elżbieta Lewandowicz** *,[†] [ID]

Institute of Geoinformation and Cartography, University of Warmia and Mazury in Olsztyn, 10-720 Olsztyn, Poland; pawel.flisek@student.uwm.edu.pl

* Correspondence: leela@uwm.edu.pl

† These authors contributed equally to this work.

**Abstract:** The aim of this study was to modify an algorithm for mapping service areas, also known as access areas. The algorithm is widely applied in network analyses. Service areas are generated based on features such as road networks and base points representing selected objects or facilities. Spatial barriers in the space between road segments are not taken into account in the process of generating service areas. Such barriers include railway lines and rivers. In this study, a methodology for generating service areas that accounts for spatial barriers was proposed by designing a dedicated tool in the ModelBuilder application in ArcGIS (ESRI) software. The ModelBuilder application has limited functionality, and the developed algorithm had to be modified. The modified algorithm was verified based on spatial data from four cities. The results produced by standard analytical methods were compared with the results generated by the modified algorithm. The study demonstrated that spatial barriers decrease the size of service areas. The modified algorithm generates more reliable results than standard methods.

**Keywords:** network analysis; service areas; spatial barriers

## 1. Introduction

Research into road accessibility dates back to more than a century ago. The first study [1] investigated the time of travel from London to any destination in the world, and it gave rise to various analytical methods for determining cost distance (distance, time of travel and fuel costs). Network analyses have found applications in numerous fields and disciplines [2]. This article focuses on transportation network analysis. Different techniques were developed for mapping access to road and service areas [3–7]. In cartography, service areas are generally presented with the use of lines known as isochrones, isograms, isolines or equidistant lines [8].

At present, routes are mapped with the use of GIS navigation tools for network analyses. These tools are also applied to solve transportation problems related to accessibility and the determination of service areas. GIS programs are equipped with analytical tools for mapping service areas [9,10]. The relevant algorithm relies on road networks [11], railway lines [12], rivers, pipelines, and electric lines. It is deployed to determine transport accessibility [13], areas of influence [14–16] and risk zones [17,18]. The results are presented in cartographic form. Service areas are determined with the use of polygons where accessibility values are preset in intervals. Such analyses are performed to plan changes in the structure of transportation networks [19] and to localize construction projects. They are carried out to evaluate the effectiveness of the proposed changes in the structure of the network and to select the optimal sites for new projects.

An algorithm for determining service areas is based on a model of the road network and base points. A base point describes the location of an object (facility) whose accessibility will be determined. Accessibility is determined in view of the adopted distance cost [20], such as the maximum distance that a client is willing to travel to reach an object (facility). In the first stage, the algorithm calculates the effective reach of the road network from a base point at a given cost [20]. It generates selected road segments that extend from the base point in a star pattern (Figure 1a). In the following stage, the algorithm generates a Triangulated Irregular Network (TIN) based on road network points (Figure 1b). The vertices of the TIN are assigned weights from the road network, which are equal to, for example, the road distance from the base point. In the TIN, linear data from the road network can be extrapolated to surface data (space between roads). TIN data are interpolated to produce service area polygons in the adopted cost intervals. Such analyses have been presented in the literature [21] in research studies addressing transportation [22], economic [23], social [13,24], security [25] and other problems [26]. The results of an exemplary analysis with a cartographic presentation of a service area are shown in Figure 2.

**Figure 1.** Generation of service areas: (**a**) the effective reach of the road network from a base point at a given cost, (**b**) a Triangulated Irregular Network (TIN) developed based on road network points.

A solution based on the road network and cadastral parcels [27] is presented in Figure 3.

Point, linear and polygon barriers on a road can be taken into account with the use of network analysis tools [28]. The road network model is modified, the shortest route is determined, and service area polygons are mapped. However, the service area is mapped only based on the road network. The space between adjacent roads is regarded as uniform. Linear barriers such as rivers and railway lines are not taken into consideration in the process of generating a service area [29]. Therefore, the results do not always accurately reflect reality, and they have to be verified.

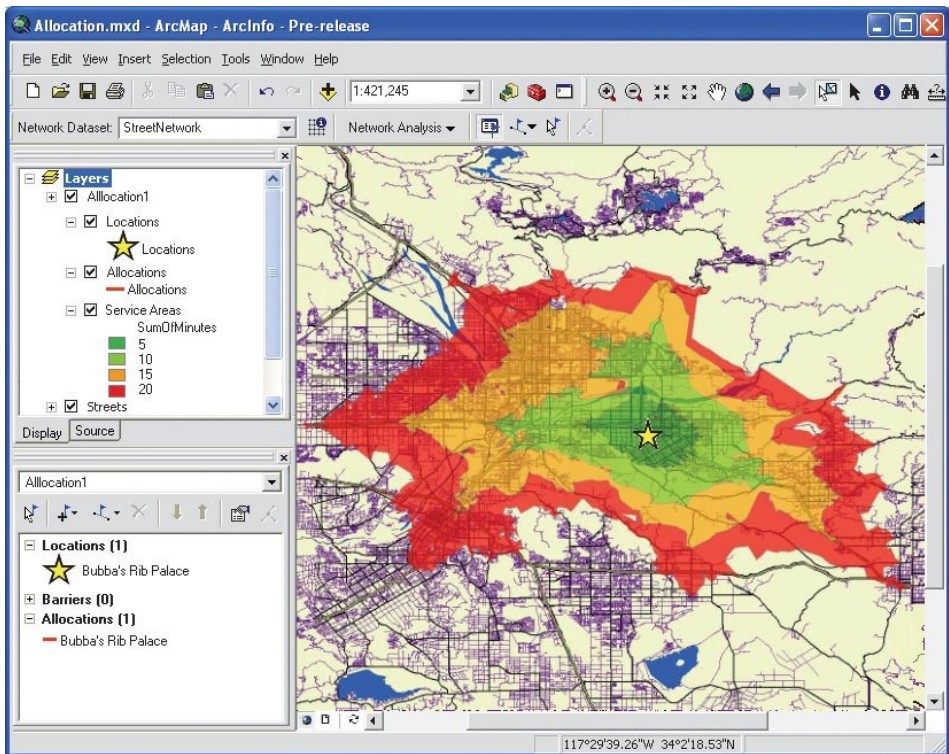

**Figure 2.** Determination of a service area around a selected base point, including polygons denoting different levels of accessibility [30].

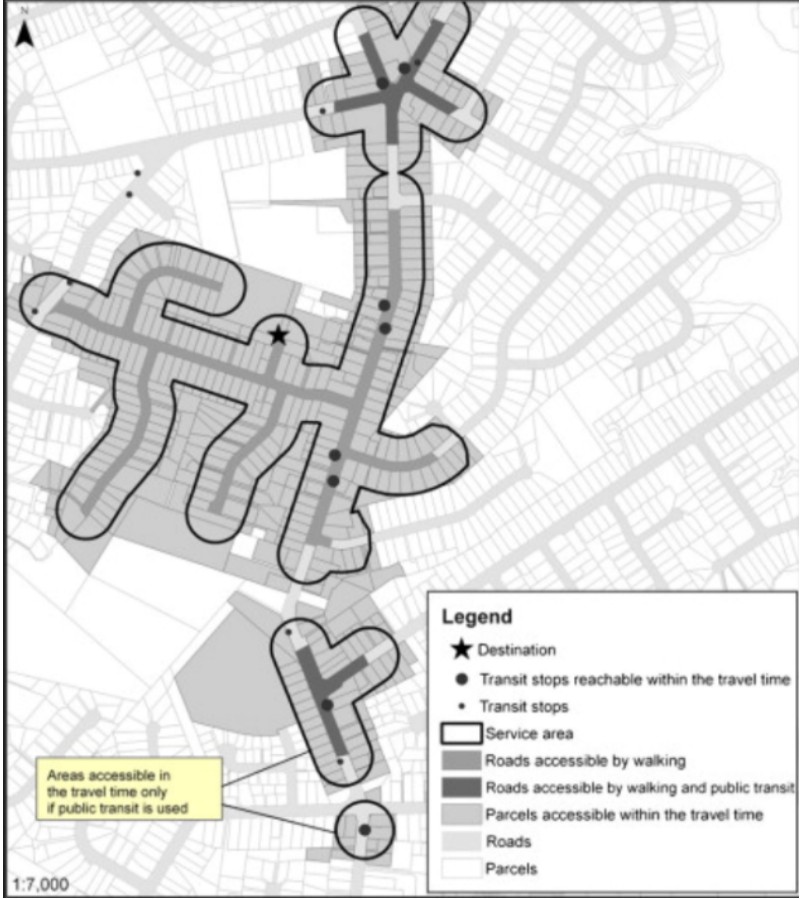

**Figure 3.** A service area mapped based on the road network and cadastral parcels [27].

Point, linear and polygon barriers can be included in the network analysis module of ArcGIS software. These barriers are taken into account in the process of developing network models (as restrictions or added costs). However, these barriers cannot be incorporated in the procedure of generating service areas. The tool for generating service areas relies only on the network model, and it does not account for certain barriers, such as river sections between bridges (which are not permeable zones). The results of the analysis performed with ArcGIS tools are presented in Figure 4. Barriers (blue lines) were taken into account in the network development process. Service area polygons were generated based only on the network model. Barriers do not appear to influence the shape of service area polygons. Barriers should denote the boundaries of service areas, but, instead, they intersect the relevant polygons. In this study, attempts were made to modify the methodology for determining service areas to account for linear barriers.

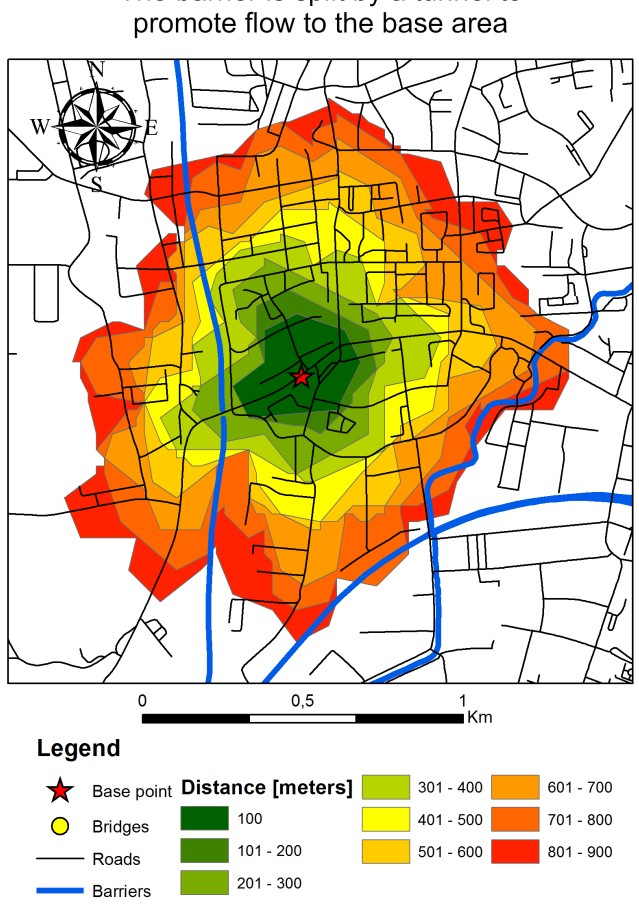

**Figure 4.** A road network developed in view of linear barriers and a service area. Service area polygons do not account for barriers.

Raster analyses are recommended for evaluating transport accessibility in view of the existing barriers [14]. Raster analyses produce reliable graphic presentations if pixel values are set correctly. The results are numerical values describing the accessibility intervals of different service areas, but they are not always highly accurate. These values are calculated based on pixels or the adopted reference units (hexagons, squares, rhombuses) (Figure 5). The results are based on the distances calculated from the vicinity of pixels. These distances are always longer than the distances calculated from vector data. The results are thus deformed. The modifiable areal unit problem (MAUP) is an important issue in spatial analyses [31]. It is one of the reasons why the vector approach was used in accessibility analyses.

In the present study, an existing vector-network analysis algorithm was modified in an attempt to generate service areas with greater accuracy. Linear barriers were taken into consideration in the modified algorithm.

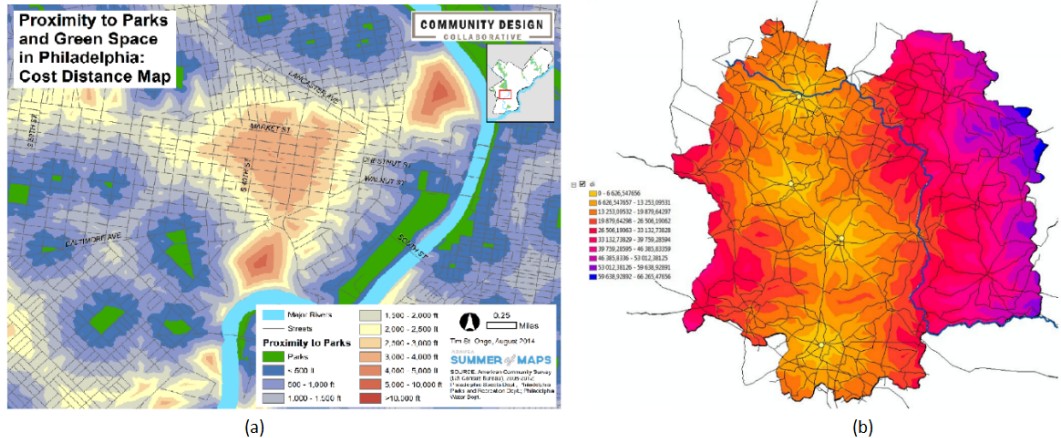

**Figure 5.** Determination of service areas in raster analysis: (**a**) [32]; (**b**) [33].

The aim of this study was to propose a methodology for modifying the algorithm for generating service areas. Barriers in the space between roads were taken into account in the modification process. The analysis was carried out based on vector data, the road network, linear barriers (rivers and railway lines) and a base point. Preliminary service areas were generated based on vector-based representation in TIN. The final result was a raster image that preserves the accuracy of vector maps (continuous). The spatial resolution of the result raster was set in the process of converting vector data to raster data. Therefore, the proposed methodology can be described as a hybrid vector–raster approach. The presented method had been initially verified in a previous study by processing spatial data manually with the use of GIS tools [34]. In the present study, attempts were made to automate the process. An algorithm for generating service areas with barriers will automate the process. The results should be free of the errors that are encountered during manual data processing. The proposed tool will support analyses of various types of objects. It will verify the algorithm for different configurations of network data and different barriers. A dedicated tool for generating service areas that accounts for linear barriers was developed with the ModelBuilder application in ArcGIS software (ESRI, Redlands, CA, USA).

## 2. Materials and Methods

### 2.1. Analyzed Area and Input Data

The algorithm was tested on spatial data relating to urban areas. The data were acquired from the Database of Topographical Objects in 1:10,000 scale (BDOT10). Four differently sized cities were selected for the analysis. Preliminary tests were conducted in the city of Olsztyn. The remaining three cities (Elbląg, Ełk and Gronowo Elbląskie) were selected for the study because their downtown areas are intersected by rivers and railway lines.The choice of the evaluated cities was dictated by the availability of the relevant data and the presence of barriers. For the needs of this analysis, cities had to be intersected by rivers and railway lines. One base point was selected in every city, and its accessibility was analyzed (Figure 6). The location of base points differed in the analyzed cities. In Ełk and Elbląg, the base points were railway stations (Figure 6a,b). In Olsztyn, the base point was the center of the university campus (Figure 6c). In Gronowo Elbląskie, the base point was situated in a downtown area next to the City Hall (Figure 6). All base points constitute important landmarks for local communities.

## Data used for analysis

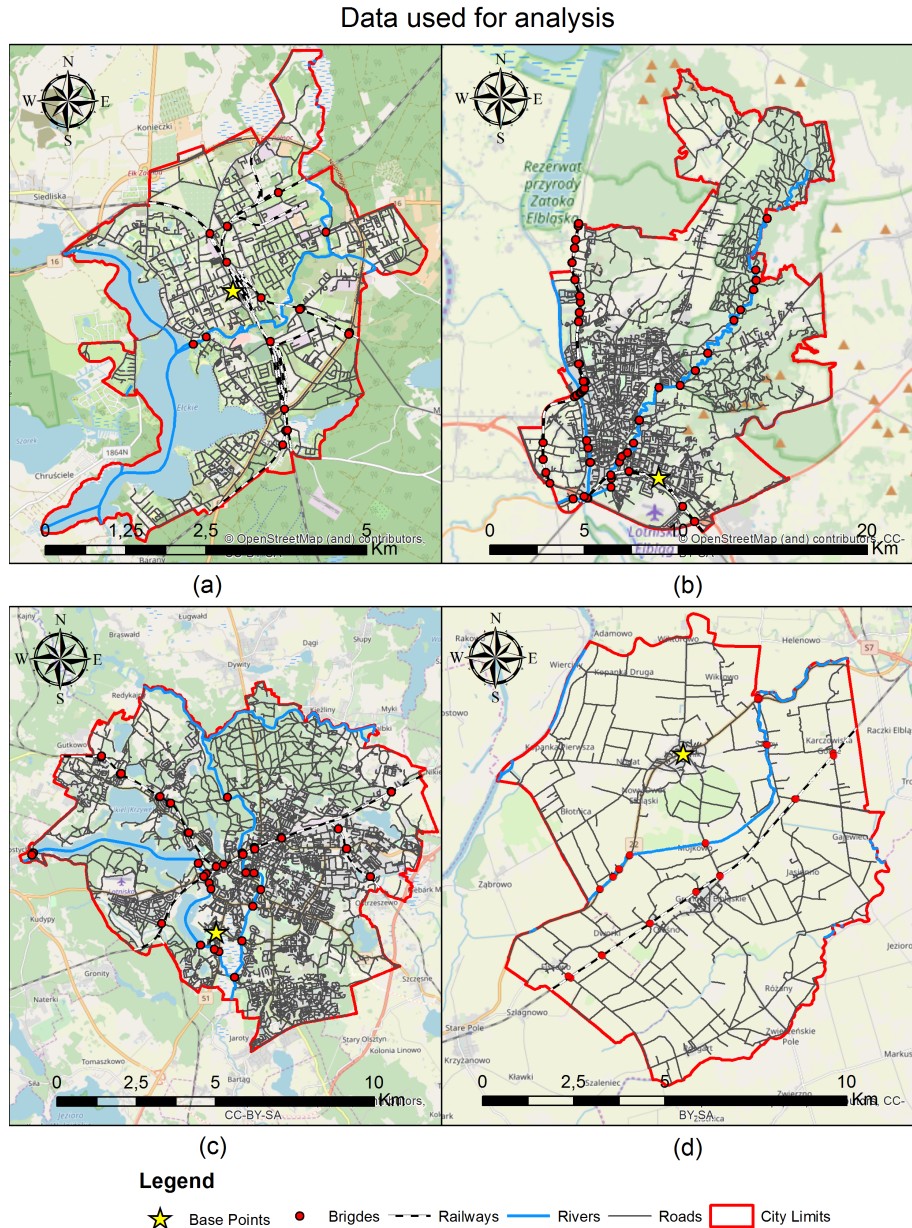

**Figure 6.** The analyzed objects with barriers (roads, rivers), bridges and base points: (**a**) Ełk; (**b**) Elbląg; (**c**) Olsztyn, (**d**) Gronowo Elbląskie.

Linear barriers were used to divide the evaluated cities into sub-areas (Figure 7). The boundaries of sub-areas were determined by intersecting the analyzed area with the adopted linear barriers. City boundaries and linear barriers for mapping sub-areas are presented in Figure 7. The data for Olsztyn are presented in Figure 7a,b; the data for Gronowo Elbląskie are presented in Figure 7c,d; the data for Ełk are presented in Figure 7e,f; and the data for Elbląg are presented in Figure 7g,h.

Process of dividing cities into sub-areas based on linear barriers

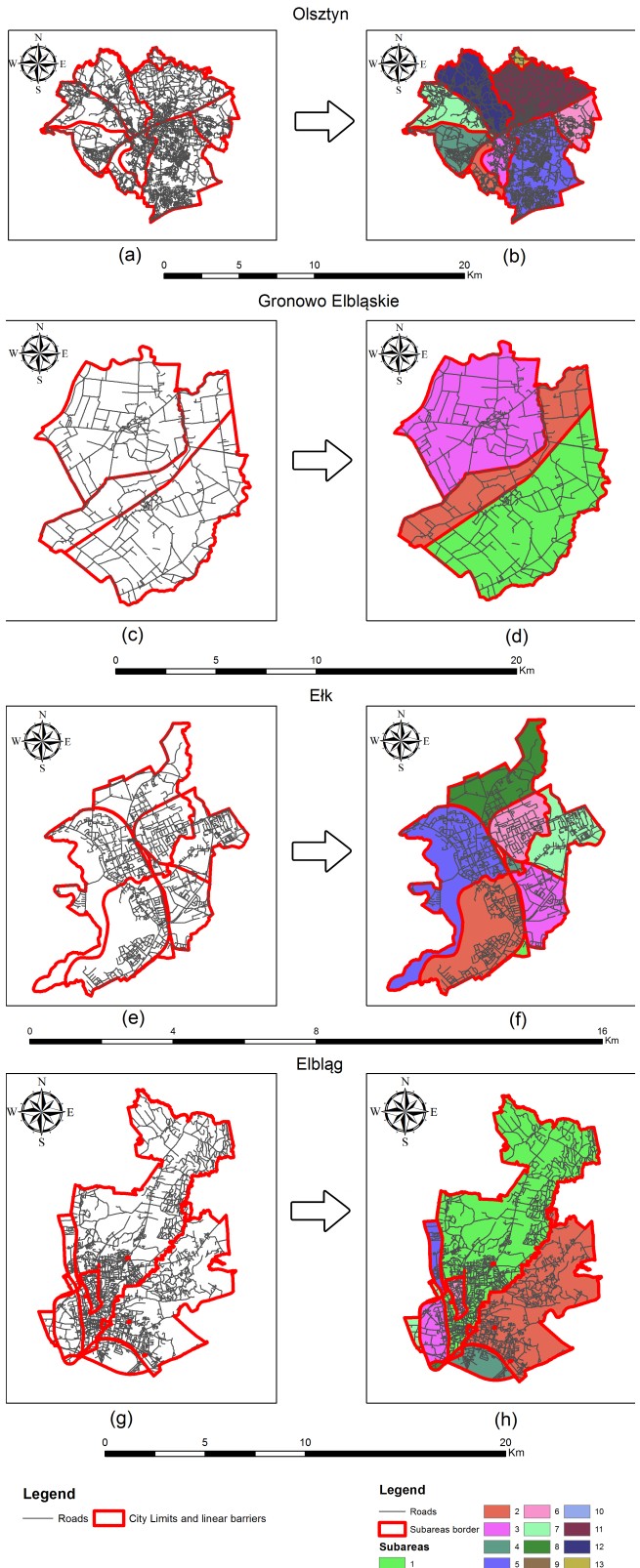

**Figure 7.** Division of the studied object into sub-areas. Linear barriers constitute the boundaries of sub-areas in every analyzed city: (**a**,**b**) Ełk; (**c**,**d**) Elbląg; (**e**,**f**) Olsztyn, (**g**,**h**) Gronowo Elbląskie.

The identified sub-areas are connected by roads and bridges. Detailed data for the analyzed cities are presented in Table 1.

**Table 1.** Characteristics of the tested objects.

| City | Area [km$^2$] | Road Length [km] | Barrier Length | Number of Sub-Areas (i) | Number of Bridges (n) |
|---|---|---|---|---|---|
| Olsztyn | 88.22 | 735.36 | 105.53 | 13 | 43 |
| Ełk | 20.89 | 179.51 | 44.66 | 8 | 18 |
| Gronowo Elbląskie | 88.89 | 244.93 | 28.30 | 3 | 30 |
| Elbląg | 79.71 | 579.40 | 56.12 | 9 | 113 |

*2.2. Methods*

The previous methodology had been described and tested in an earlier study [34]. The previously applied methodology was modified in this article. The introduced changes are apparent already at the beginning of the modeling process. In the previous study, barriers were taken into account already at the network modeling stage when the road network was divided into segments in the identified sub-areas. Sets of network data were then built in sub-areas. Service areas were generated separately in every sub-area.

The present methodology relies on one set of network data covering the entire city. Barriers are taken into account during the generation of service areas. The new methodology and the relevant formulas are described below.

The method accounts for the analyzed area $A$, road network $R$, linear barriers $B = B_k$ and a base point $U$. The aim of the analysis was to determine the service area for base point U.

Barriers $B$ divide the analyzed area $A$ into sub-areas $A_i$ (1, 2). Bridges, overpasses and crossings constitute points $BR$ which connect sub-areas, and they are an important element of the road network $R$ (3). Points $BR$ guarantee the continuity of the road network $R$. As a result, the road network aggregates sub-areas Ai of area $A$ into a cohesive structure:

$$A \cap B = \{A_1, A_2, A_3, ..., A_i\}, \tag{1}$$

$$\{A_1, A_2, A_3, ..., A_n\} \subset A, \tag{2}$$

$$R \cap B = \{BR_1, BR_2, BR_3, ..., BR_n\}, \tag{3}$$

where $n$ is the number of bridges.

Every bridge $BR_n$ was assigned a weight $W_{BR_n}$ denoting the minimal cost of travel from base point $U$ to bridge $BR_n$. In the algorithm, bridge weights $W_{BR_n}$ represented the smallest possible distance between $U$ and $BR_n$ in the road network (4). The shortest route was identified to maximize accuracy. In the previous approach [34], the weight of a bridge was determined based on the values from the neighboring sub-area:

$$W_{BR_n} = min(|U, BR_n| \in R), \tag{4}$$

where $n = 1, 2, 3, ..., n$

Base point $U$ is located in sub-area $A_i$ referred to as the base area $A_b$. It was assumed that $A_b = A_1$ (5, 6):

$$U \in A_b, \tag{5}$$

$$A_b \subset A \text{ and } A_b = A_1. \tag{6}$$

Service area $SA_U$ was mapped to determine the accessibility of base point $U$ in area $A$. In the algorithm, service sub-areas $SA_i$ were determined in successive sub-areas $A_i$. In the first step, service area $SA_1$ was mapped in base area $A_1$. In successive steps, service areas were determined in every sub-area $SA_i$, where $i \neq 1$. Service areas were mapped from every bridge located on the boundary of sub-area $SA_i$ to produce a set of preliminary service areas $SA_i^{BR_r}$, where $r$ denotes the number

of bridges on the boundary of sub-area $A_i$ (7). Vector data relating to $SA_i^{BR_r}$ (8) were converted to raster data $RSA_i^{BR_r}$(8, 9). The pixels in the raster image of service area $RSA_i^{BR_r}$ were assigned $SA_i^{BR_r}$ values that are equal to $W_{SA_i}^{BR_r}$. In raster images of service areas $RSA_i^{BR_r}$, pixel values $W_{SA_i}^{BR_r}$ were adjusted to account for bridge weights $W_{BR_n}$. This operation produced new pixel values in raster image $W_{SA_i}^{BR_r^C}$ (10, 11) as well as modified sets of service areas $RSA_i^{BR_r^C}$ (12). In successie steps, raster data in sub-areas were merged by adopting the minimal values of pixels from an $r$ number of raster images $RSA_i^{BR_r^C}$ to obtain $SA_i^R$ (13). In the following step, all data from successive sub-areas $SA_i^R, i = 1, 2, 3, ..., i$ were merged (14). The result raster $SA_U^R$ presents the service area of base point $U$ that accounts for linear barriers $B$:

$$\{BR_i^{BR_1}, BR_i^{BR_2}, BR_i^{BR_3}, ..., BR_i^{BR_r}\}, \tag{7}$$

$$BR_i^{BR_r} \xrightarrow{\text{conversion}} RSA_i^{BR_r}, \tag{8}$$

$$\{RSA_i^{BR_1}, RSA_i^{BR_2}, RSA_i^{BR_3}, ..., RSA_i^{BR_r}\}, \tag{9}$$

where: $i$ = number of sub-areas, $i = 1, 2, 3, 4, ..., i$
$r$-number of bridges on the boundary of area $SA_i$, where $r < n$,

$$W_{SA_i}^{BR_r^C} = W_{SA_i}^{BR_r} \cup W_{BR_r}, \tag{10}$$

$$RSA_i^{BR_r} \xrightarrow{\text{adjustment}} RSA_i^{BR_r^C}, \tag{11}$$

$$\{RSA_i^{BR_1^C}, RSA_i^{BR_2^C}, RSA_i^{BR_3^C}, ..., RSA_i^{BR_r^C}\}, \tag{12}$$

where: $i = 1, 2, 3, ..., n$

$$\{RSA_i^{BR_1^C}, RSA_i^{BR_2^C}, RSA_i^{BR_3^C}, ..., RSA_i^{BR_r^C}\} \xrightarrow{\text{mininum association}} SA_i^R, \tag{13}$$

where: $i = 1, 2, 3, ..., n$

$$SA_U = SA_1^R + SA_2^R + SA_3^R + ... + SA_i^R. \tag{14}$$

The presented method was developed manually with the use of GIS tools. It should be noted that the same operations were iterated multiple times in successive sub-areas. The developed algorithm was saved in the ModelBuilder application to automate analyses in other objects.

*2.3. Algorithm*

The first approach to implementing the proposed methodology was based on the method developed in a previous study [34]. The manual procedure could not be incorporated into the algorithm in the ModelBuilder tool. The algorithm would have to select successive sets of network data representing the road network in sub-areas. Service areas in sub-areas should be mapped based on these data sets. The base points for determining service areas in sub-areas should also include bridges. This solution was difficult to implement in the ModelBuilder tool. For example, the information about which bridge points are located in a given sub-area would have to be available to the algorithm. This solution was abandoned, and the existing algorithm was modified. As a result, numerous sets of network data did not have to be developed. In the adopted approach, a single set of network data was determined to cover the entire network. The boundaries of sub-areas were taken into account in the process of generating service areas in sub-areas. The process of dividing the analyzed area into sub-areas with the use of the algorithm developed in the ModelBuilder application is presented in Figure 6.

The algorithm relies on the following data inputs: the road network $R$, base point $U$, bridges $BR$ and barriers $B$. The modified algorithm differs only in that it accounts for linear barriers. Linear barriers were adopted as the boundaries of sub-areas; therefore, the boundaries of sub-areas were composed of the segments of boundaries of the evaluated object ($BA$), and barriers (rivers, railway lines). In this approach, barriers constitute the boundaries of sub-areas, i.e., the boundaries of object polygons $A_i$. Barriers $BA_i$ and preliminary access areas were determined in the process of generating boundaries $A_i$.

In the presented approach, the main part of the algorithm consists of a single iterator. The iterator operates on a set of points ($P$) composed of base point $U$ and bridge points $BR$, $P = \{U, BR_1, BR_2, \ldots, BR_n\}$. In the first step, the iterator calculates weights $W_{BR_n}$ for every point in set $P$ on the assumption that weight $U = 0$. Preliminary service areas are determined based on set $P$. A singe service area is mapped from point $U$. The number of service areas mapped from points $BR_n$ is determined by the number of sub-area boundaries with point $BR_n$. In most cases, bridges connect two sub-areas, but bridges intersecting two barriers—a railway line and a river—can connect four sub-areas. The boundaries of sub-areas (and barriers) are taken into account in the process of mapping service areas. The process of generating sets $SA_i^{BR_r}$ is presented in a flow chart in Figure 8.

Successive parts of the algorithm were calculated with the use of Formulas (8)–(15). Vector data were converted to raster data (8, 9), pixels in service areas were assigned weights (10, 11, 12), and service areas mapped from successive bridges in sub-areas were aggregated (13) based on minimal values. In the final step, the results from sub-areas (13) were merged in a single raster image (14). The raster image presents the service area around base point $U$ in view of linear barriers $BR_n$.

## 3. Results

The method developed in the ModelBuilder application was used to determine service areas in the analyzed objects. The results were presented graphically in two variant solutions. In the first variant, service areas were identified based only on the road network. This is the standard ESRI approach. In the second variant, the proposed methodology was used to develop an algorithm that accounts for linear barriers such as rivers and railways. The results of the analysis performed in the cities of Ełk (Figure 9), Gronowo Elbląskie (Figure 10), Olsztyn (Figure 11) and Elbląg (Figure 12) are presented below.

The results of the analysis are presented graphically in Figures 9–12. The results generated with the use of the ArcGIS tool indicate that service area polygons intersect barriers. Barriers do not obstruct the generation of service areas. The results generated by the proposed methodology indicate that barriers influence the shape of service area polygons. This is particularly apparent in locations where longer barrier sections are accompanied by very few bridges. When the number of bridges is high, the results are consistent with those obtained in the standard approach.

The differences in the results produced by the standard approach and the proposed methodology are particularly visible in Ełk (Figure 9). The base point was located in the railway station, on one side of the railway track. The railway station is more accessible in the standard approach than in the developed methodology. Railway tracks obstruct access to the railway station from the other side of the track. The city of Ełk is characterized by long river sections without any bridges. The compared methods thus generate different results relating to accessibility.

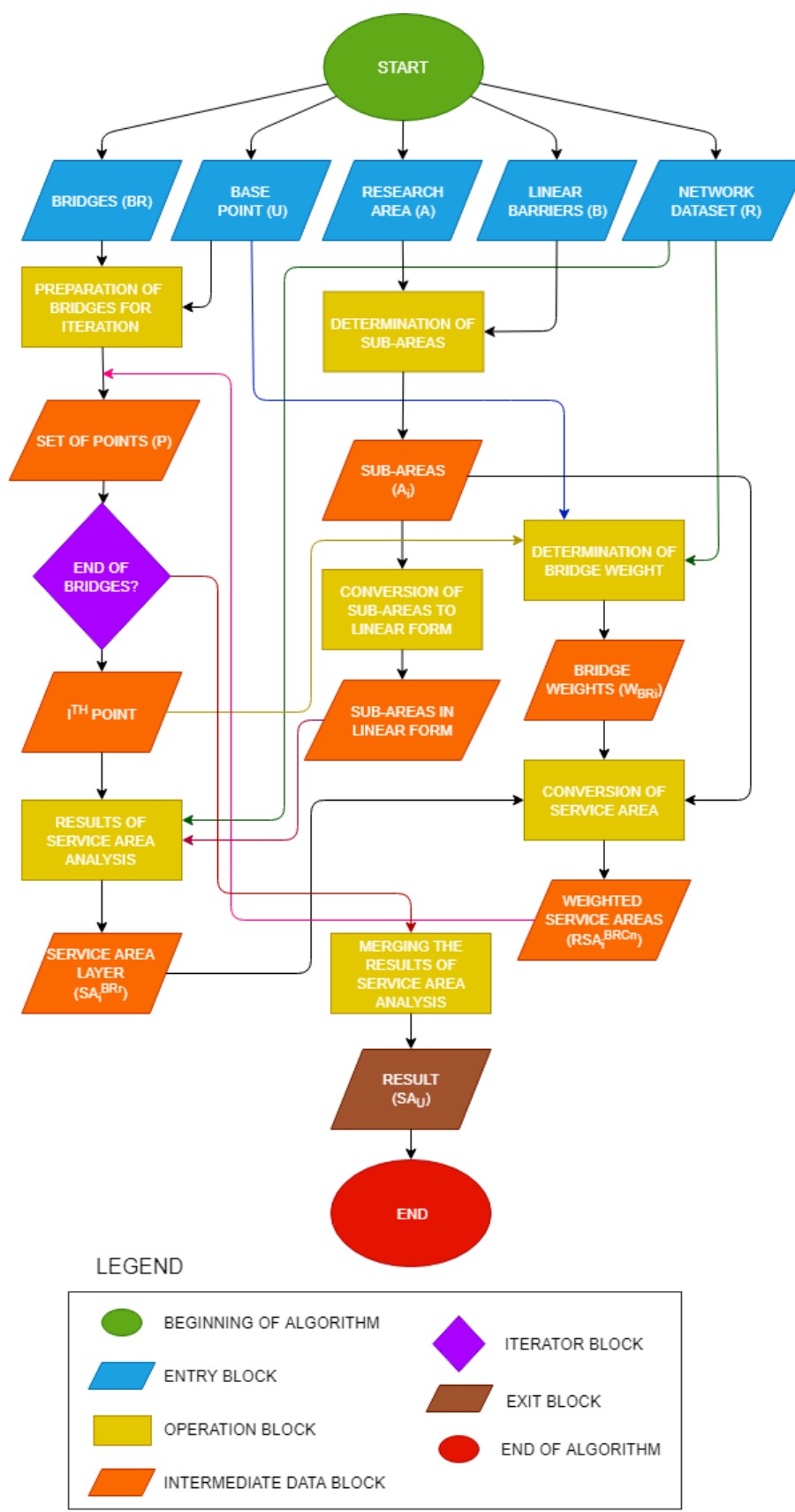

**Figure 8.** Data processing flow chart in the ModelBuilder application for determining service areas from a single base point in view of linear barriers.

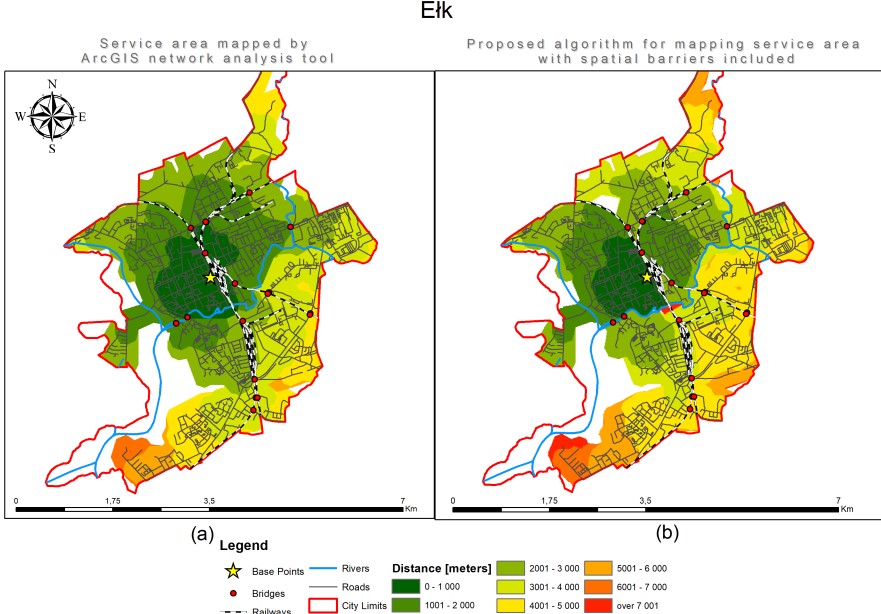

**Figure 9.** Service areas identified in the city of Ełk: (**a**) results generated by the standard network analysis tool in ArcGIS; (**b**) results generated by the algorithm developed in the ModelBuilder application in ArcGIS.

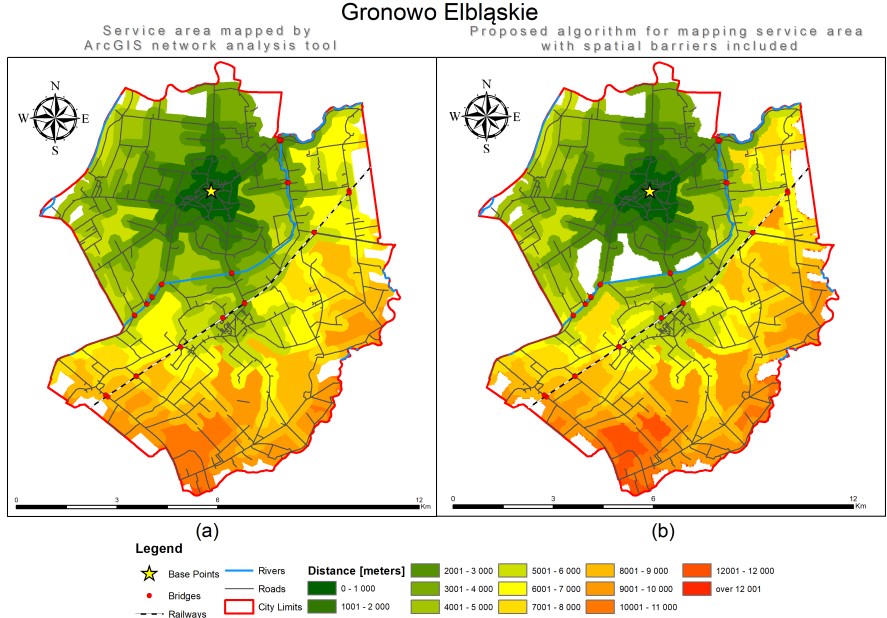

**Figure 10.** Service areas identified in the city of Gronowo Elbląskie: (**a**) results generated by the standard network analysis tool in ArcGIS; (**b**) results generated by the algorithm developed in the ModelBuilder application in ArcGIS.

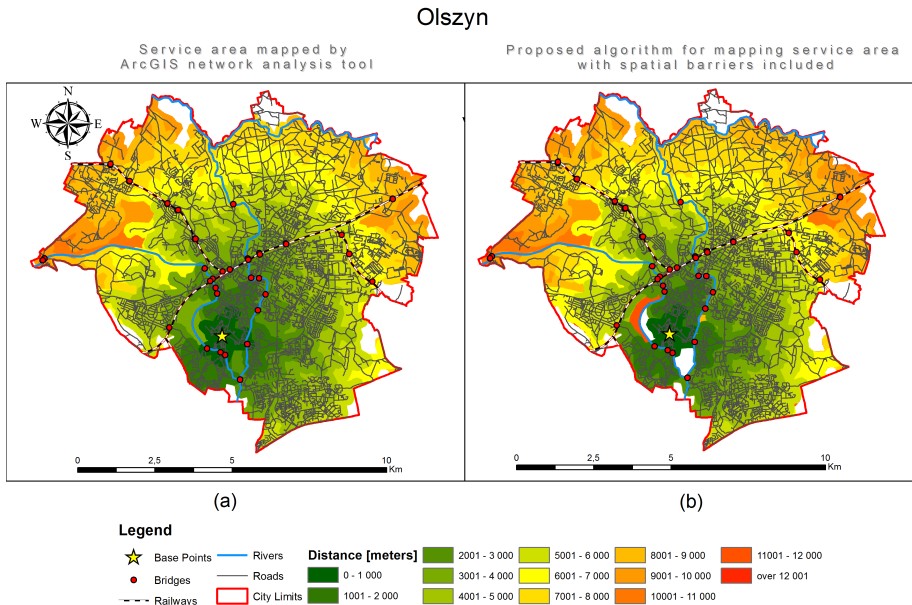

**Figure 11.** Service areas identified in the city of Olsztyn: (**a**) results generated by the standard network analysis tool in ArcGIS; (**b**) results generated by the algorithm developed in the ModelBuilder application in ArcGIS.

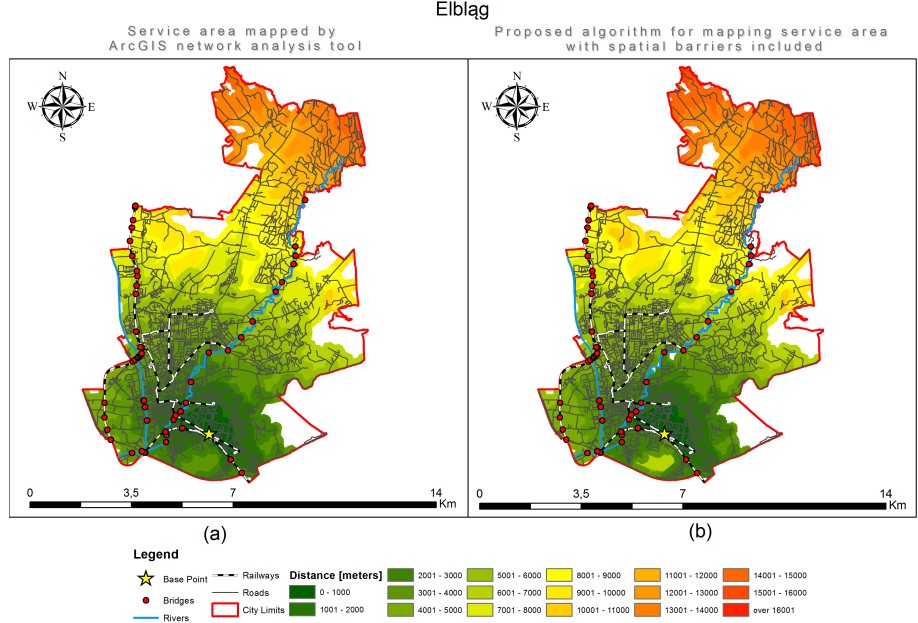

**Figure 12.** Service areas identified in the city of Elbląg: (**a**) results generated by the standard network analysis tool in ArcGIS; (**b**) results generated by the algorithm developed in the ModelBuilder application in ArcGIS.

In Gronowo Elbląskie, the base point is located further away from barriers. The road network is distributed perpendicularly to the barriers. Both methods induce only minor changes in the shape of service area polygons. Differences are noted only southeast of the base point, and they span the length of around one kilometer. The differences in the results for the city of Olsztyn are minimal due to the presence of an extensive road network and numerous bridges. The only noticeable changes

can be found in the northeastern and southwestern part of the city where bridges are absent. Similar differences are observed in Elbląg, which has fewer bridges.

Polygons characterized by different accessibility in the adopted intervals are relatively regularly distributed from the base point. Polygons were more regularly distributed in the solutions generated by the standard approach. Polygons were less regularly distributed when the proposed method that accounts for barriers was implemented. In many cases, linear barriers denoted polygon boundaries. The influence of barriers is visible in longer segments without bridges, in particular in the vicinity of a dense road network. A high number of bridges minimize the influence of linear barriers on the identified service areas.

The results were presented graphically in Figures 13–16. The distribution of variously-sized service areas was determined within the adopted cost distance intervals. Blue lines based on Euclidean distance were added to the diagrams to denote the size of service areas determined from base points based on buffers. Blue lines were introduced as a reference to facilitate the interpretation of results. The boundaries of the analyzed area and the location of base points are responsible for the breakpoints on the blue line. In Figure 17, the blue line was mapped based on the data for the city of Olsztyn.

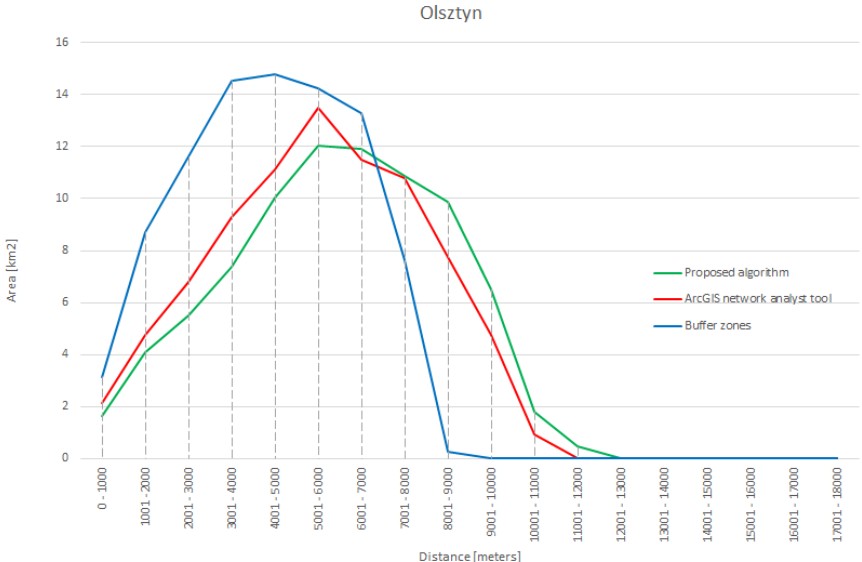

**Figure 13.** The results generated by different methods for identifying service areas in the city of Olsztyn.

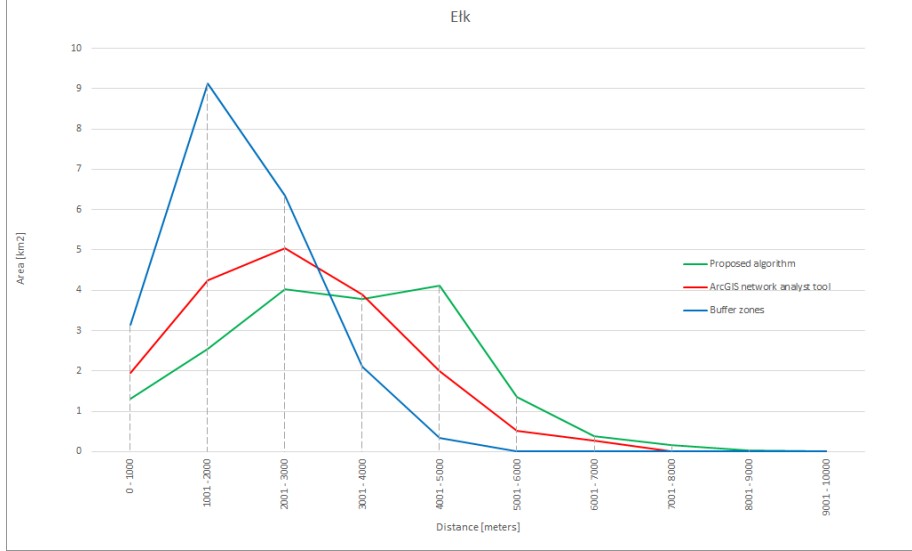

**Figure 14.** The results generated by different methods for identifying service areas in the city of Ełk.

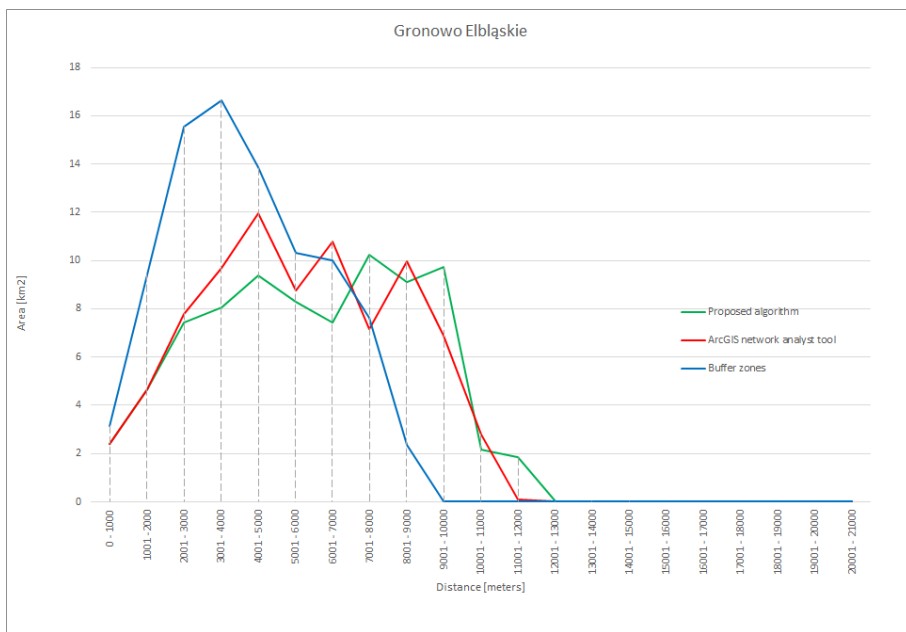

**Figure 15.** The results generated by different methods for identifying service areas in the city of Gronowo Elbląskie.

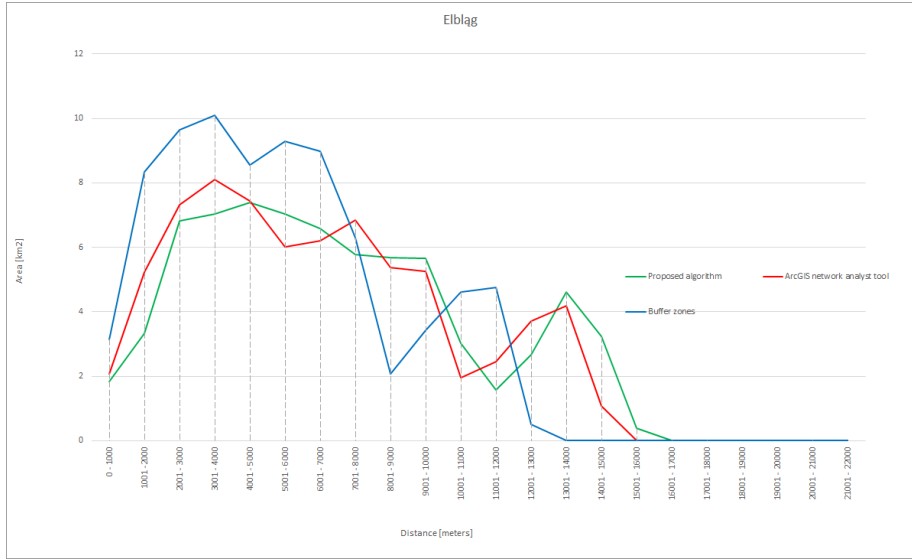

**Figure 16.** The results generated by different methods for identifying service areas in the city of Elbląg.

The presented lines have a parabolic shape which supports the determination of the maximum values. Their parabolic shape can be attributed mainly to the irregular boundaries of area *A* and the location of base point U. The lines presenting the results of the analysis conducted with the use of the proposed methodology are flattest and most elongated. The above indicates that service areas determined based on a given area are smaller than the areas determined with the standard method or the buffer method. The service area determined in the proposed method is smallest because linear barriers decrease accessibility. The green line covers the longest distance on the horizontal axis. The service area around a point located on the boundary of the analyzed area is characterized by a higher cost distance.

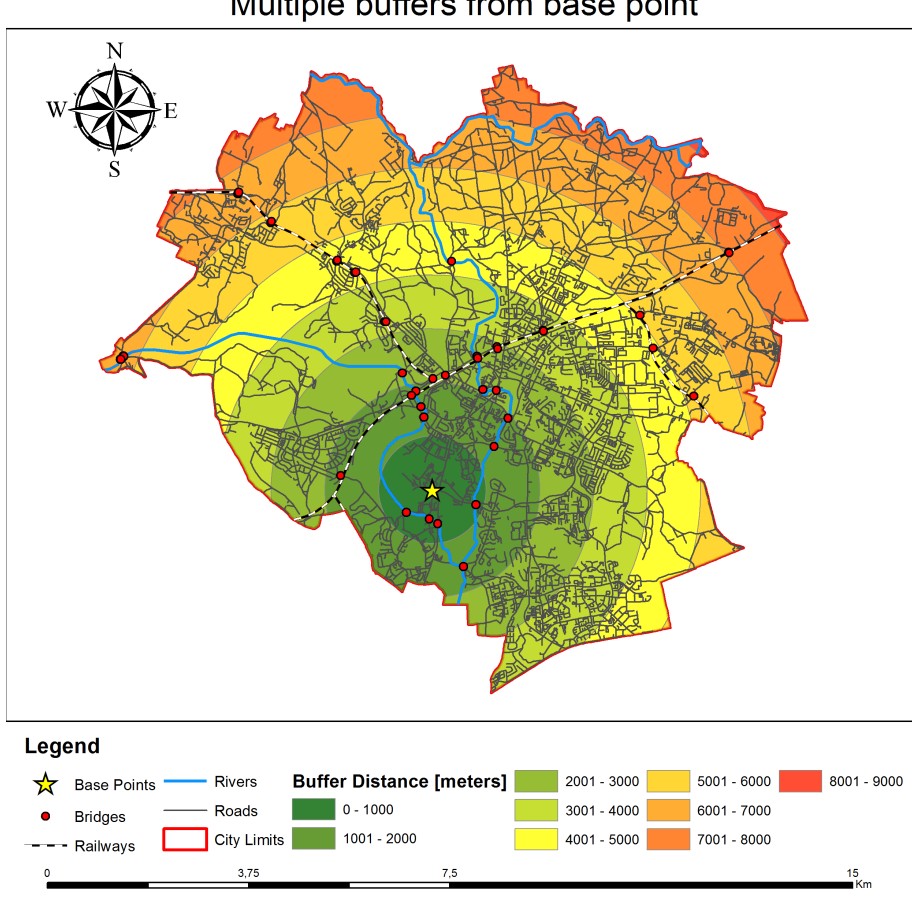

**Figure 17.** Service areas identified with the use of buffers based on Euclidean distance.

The lines mapped for the city of Olsztyn are different within the distance interval of 4–6 km. Maximum values are noted in these intervals. In Figure 14, service area buffers are cropped by the boundary when the distance exceeds 6 km. Buffer area decreases in successive distance intervals. Cropped buffers have minimal area when distance exceeds 9 km. Similar correlations were noted in service areas. The largest buffer zones were noted within the distance interval of 5 to 6 km. The green line is displaced relative to the red line, which indicates that the service area identified with the proposed methodology was smaller than that determined in the standard approach. The difference is around 1.5 km$^2$, and it increases to 2 km$^2$ within the distance interval of 9 to 10 km.

In Gronowo Elbląskie (Figure 16), the red line overlaps the green line within the distance interval of up to 3 km. Linear barriers are not observed in this interval. The influence of linear barriers becomes apparent when the distance from the base point exceeds 3 km (the red line and the green line diverge). The service areas determined with the use of the proposed method are smaller.

The differences between the service areas generated with the proposed tool and the ArcGIS tool are presented in Figure 18. Service area polygons differ considerably in every sub-area. The greatest differences can be observed in the vicinity of barriers. These differences have positive values, which indicates that the accessibility determined with the use of the developed tools is smaller.

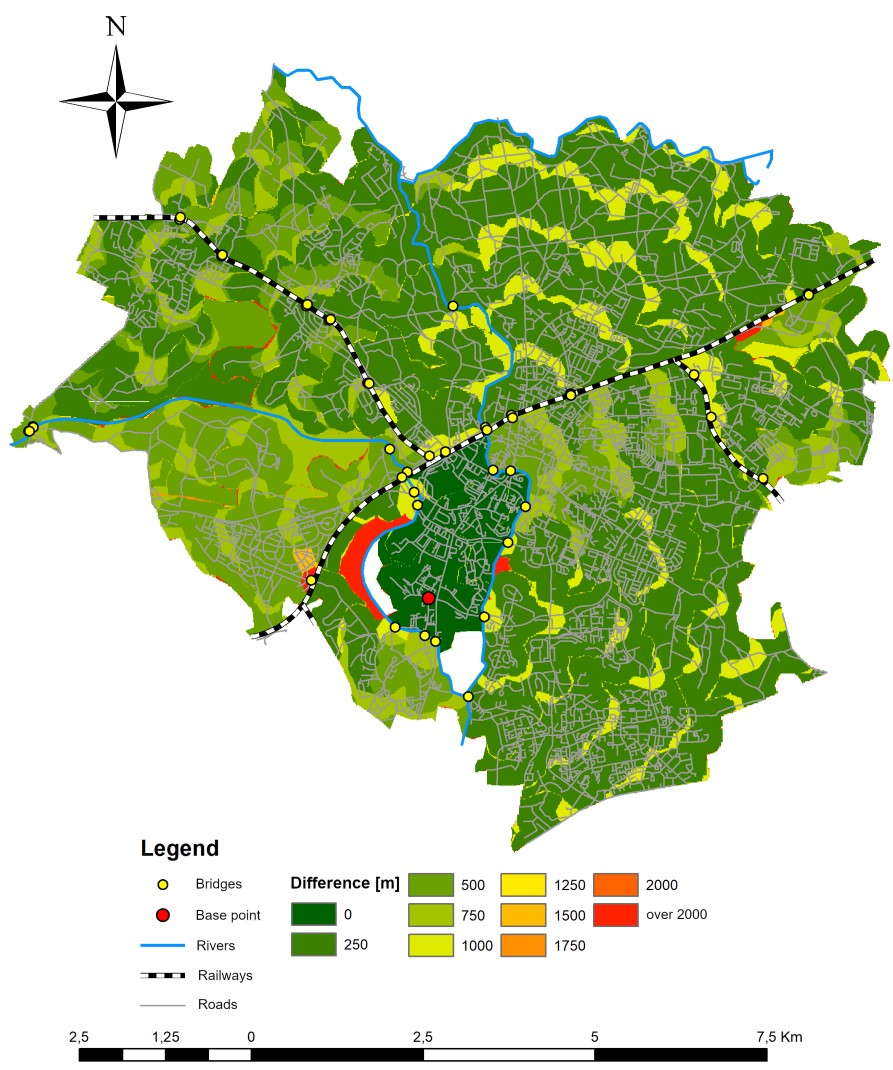

**Figure 18.** Differences in the generated values of service areas based on Olsztyn data. A differential raster based on the results generated by the proposed algorithm and the ArcGIS tool (Network Analysis).

A differential raster was generated to compare the results obtained with the use of the presented method and the manual processing method described in a previous study [33]. The resulting differences are presented in Figure 19. Base areas are characterized by the same accessibility, but the values determined in sub-areas situated remotely from base points are different. These variations could be attributed to the fact that several bridges were omitted in the manual method. In the proposed solution, the results are processed automatically, which increases their reliability. Accessibility decreases when barriers are taken into account.

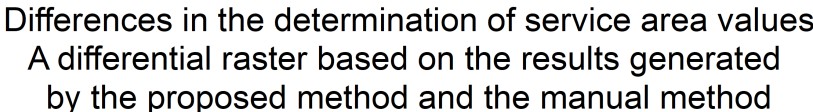

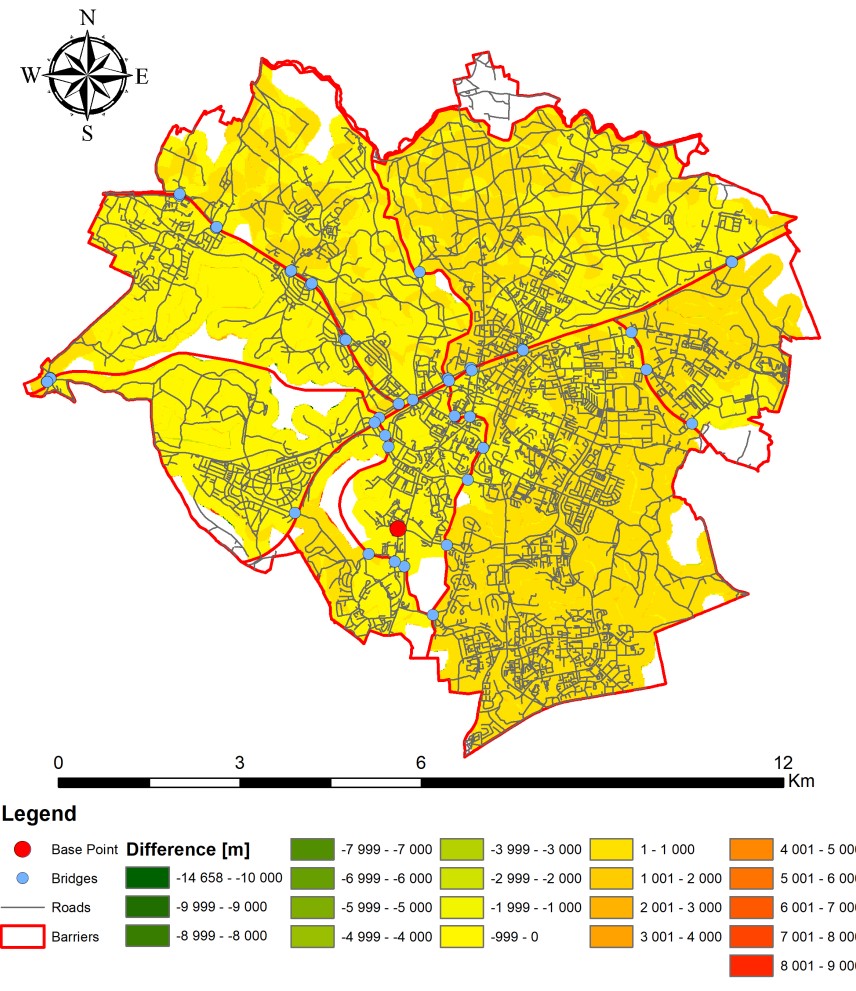

**Figure 19.** A differential raster of the values of service area polygons generated with the proposed algorithm and the manual method [34] based on Olsztyn data.

## 4. Discussion

The designed algorithm accounted for the influence of linear barriers in the process of generating service areas, which increased the reliability of the results. The algorithm was also tested on special cases where artificial networks and barriers were created. The first case involved a meandering river (Figure 20a). The construction of a bridge led to changes in service areas (Figure 20b). It should be noted that the algorithm accounted for the shortest distance from the base area to the neighboring area $A_2$ and back to the base area. In the adopted algorithm, the division of the analyzed area $A$ into sub-areas $A_i$ was not a limiting factor. The methodology developed in ArcGIS was used to determine service areas for the same sets of data. The results are presented in Figure 20c,d. Barriers were taken into account only in the proposed methodology.

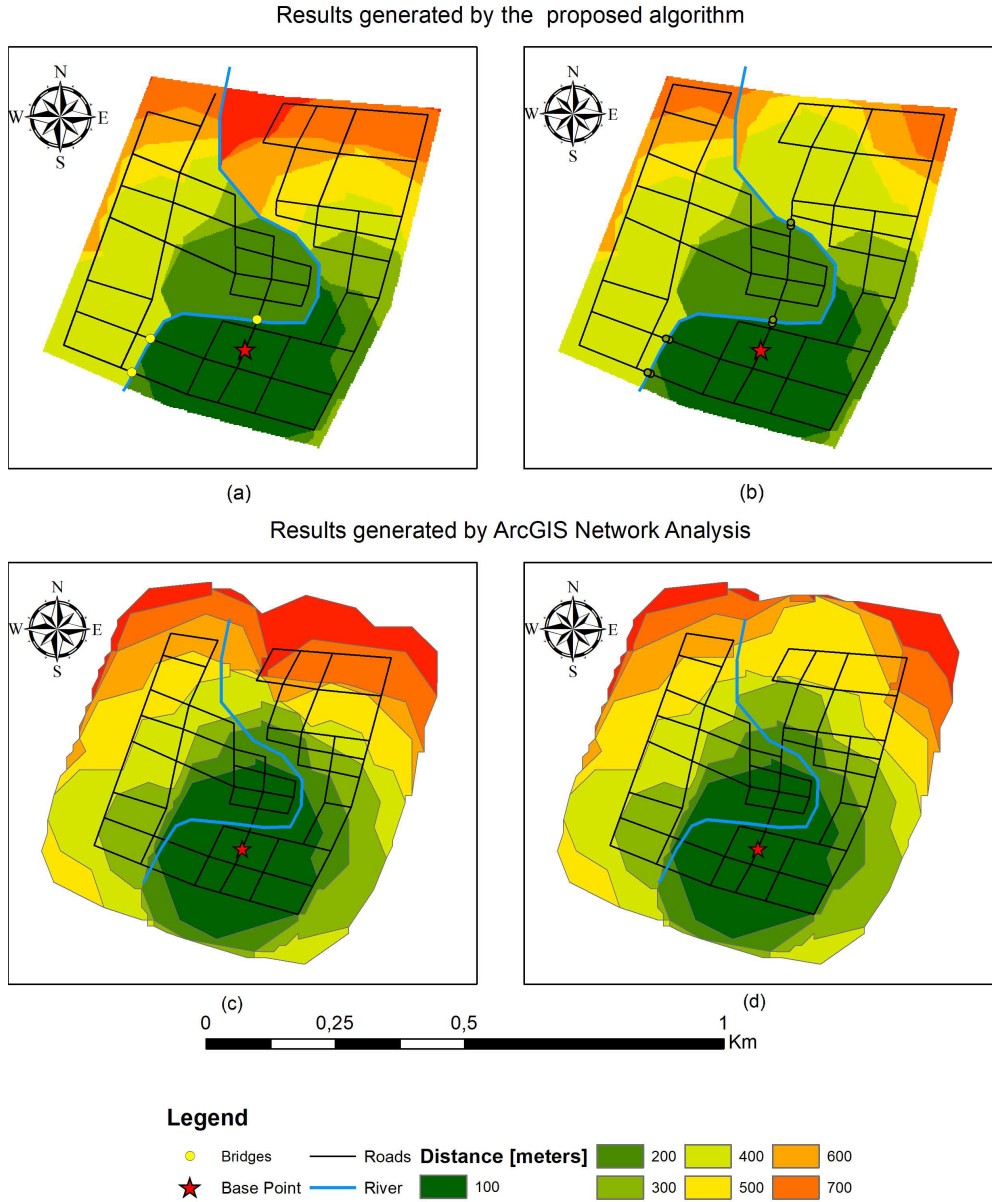

**Figure 20.** Special case of a linear barrier—a meandering river. The analyzed area was not limited to sub-areas: (**a**) results generated by the algorithm for three bridges; (**b**) results generated by the algorithm for four bridges; (**c**) results generated by ArcGIS network analysis without possible flow to base area; (**d**) results generated by ArcGIS network analysis with possible flow to base area.

In the second case, the barriers were a river and its floodplain lake (Figure 21). The lake was incorporated into the algorithm as a sub-area without a road network. The analysis was limited to sub-areas with a road network. This indicates that the scope of the analysis can be expanded by modeling barriers, bridges and excluded areas. The algorithm is capable of processing such data.

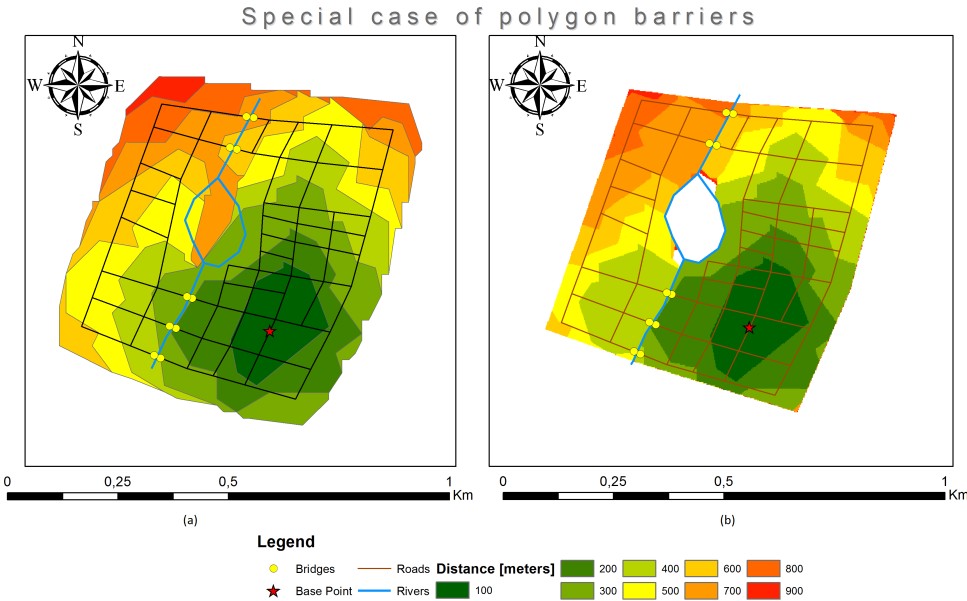

**Figure 21.** The results of an analysis for excluded areas: (**a**) ArcGIS network analysis; (**b**) proposed algorithm.

The final service areas mapped with the ArcGIS tool and the proposed methodology are presented in Figure 22. The presented example makes a reference to Figure 4 in the Introduction.

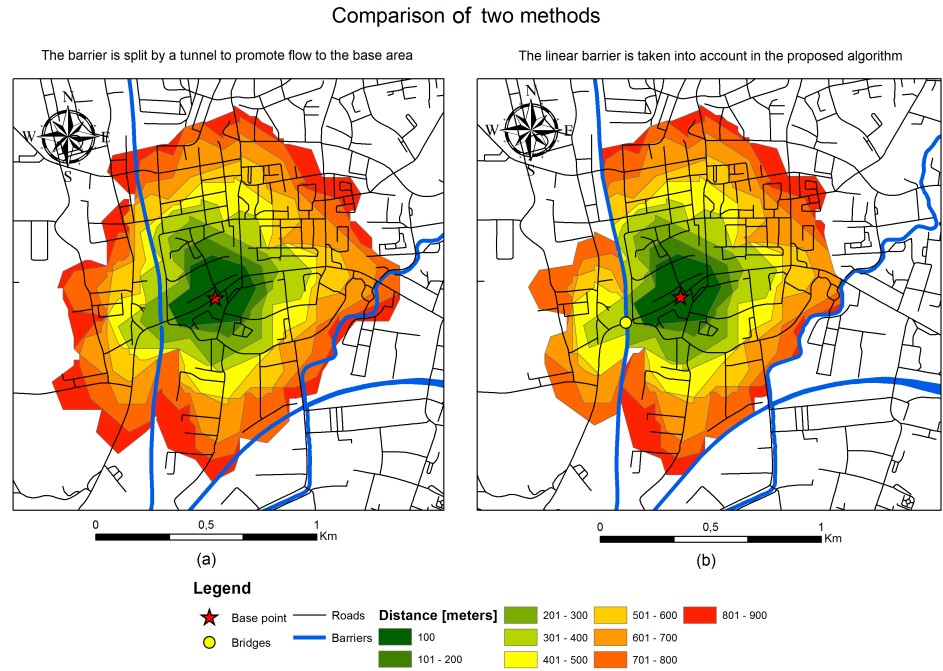

**Figure 22.** Service areas mapped with: (**a**) ArcGIS tool; (**b**) the proposed methodology for one bridge.

The effects of the proposed methodology are clearly visible in Figure 22. Barriers constrain the results of the analysis. Barriers constitute the boundaries of service area polygons. The proposed methodology generates higher-quality results than the standard approach.

## 5. Conclusions

The algorithm for generating service areas was modified to account for linear barriers (rivers, railway lines). The results generated by the modified algorithm that was implemented in the ModelBuilder tool are more reliable. The manual method failed to produce fully reliable results. Automation eliminates the errors that occur during manual processing. The ModelBuilder tool supported analyses of various datasets. In the future, the automated procedure can be used to incorporate a higher number of barriers, including buildings, retaining walls and ditches, into the model.

The process was successfully automated with the use of the ModelBuilder application. The algorithm was modified for implementation in ModelBuilder. However, automation significantly prolonged the modification process. Further research is needed to implement the algorithm with the use of programming tools, such as Python.

**Author Contributions:** Conceptualization, Paweł Flisek and Elżbieta Lewandowicz; methodology, Paweł Flisek and Elżbieta Lewandowicz; formal analysis, Paweł Flisek; writing—original draft preparation, Paweł Flisek and Elżbieta Lewandowicz; writing—review and editing, Paweł Flisek and Elżbieta Lewandowicz; visualization, Paweł Flisek

**Funding:** This research was financed as part of a statutory research project of the Faculty of Geodesy, Geospatial and Civil Engineering of the University of Warmia and Mazury in Olsztyn, Poland, entitled "Geoinformation from the theoretical, analytical and practical perspective" (No. 28.610.033300, timeline: 2017–2020).

**Conflicts of Interest:** The authors declare no conflict of interest.

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
