# Peer review of "A Methodology for Generating Service Areas That Accounts for Linear Barriers"

_ijgi, doi:10.3390/ijgi8090423_

Round 1

Reviewer 1 Report

Thank you for providing the model

Author Response

RESPONSE TO REVIEWER 1

Point 1: Thank you for providing the model

Response 1: Thank you for a positive review of our manuscript.

Reviewer 2 Report

This is an interesting paper that is modifying the algorithm for generating service areas, more precise an existing vector-network analysis algorithm is modified in an attempt to generate service areas with greater accuracy. I have a few comments:

The MS should be proofread by a native speaker – there are in the text expressions such as: The aim of the this study – Line 73. – or lack of punctuation Line 82 - the process.An . etc.

The methodology should be revised in terms of details – for example in line 111 the  2.2 methods section starts with The previous methodology had been described and tested in the study [29]. _ What should the reader understand from this sentence. This study – other study? Which previous methodology? The authors should try to give more precise details.

If the authors use the network analysis as keyword – they should make the distinction in the introduction or methodology between the transportation / road networks and other types of network approaches (SNA) that are now used in the scientific literature to describe and investigate the collaboration in environmental governance for example: A network approach for understanding opportunities and barriers to effective public participation in the management of protected areas, Soc. Netw. Anal. Min., 8 (2018), p. 31, 10.1007/s13278-018-0509-y.

Minor comments:

Figure 21 is left in red and in polish I suppose. Is it the same figure with figure 20?

Please keep on a single type of reference of figure in the ms – either fig. or figure.

Author Response

RESPONSE TO REVIEWER 2

Point 1: This is an interesting paper that is modifying the algorithm for generating service areas, more precise an existing vector-network analysis algorithm is modified in an attempt to generate service areas with greater accuracy. I have a few comments:

Response 1: Thank you for positive remarks and helpful comments that enabled us to improve the quality of our manuscript.

Point 2: The MS should be proofread by a native speaker – there are in the text expressions such as: The aim of the this study – Line 73. – or lack of punctuation Line 82 - the process.An . etc.

Response 2: The final version of the manuscript has been proofread by a native English speaker, and the relevant corrections have been made – please see the attached language editing certificate.

Point 3: The methodology should be revised in terms of details – for example in line 111 the  2.2 methods section starts with The previous methodology had been described and tested in the study [29]. _ What should the reader understand from this sentence. This study – other study? Which previous methodology? The authors should try to give more precise details.

Response 3: The previous methodology is the methodology applied in our previous study (reference No. 29 cited at the end of the above sentence), which includes linear barriers. In the present study, the entire process has been modified and automated.

Point 4: If the authors use the network analysis as keyword – they should make the distinction in the introduction or methodology between the transportation / road networks and other types of network approaches (SNA) that are now used in the scientific literature to describe and investigate the collaboration in environmental governance for example: A network approach for understanding opportunities and barriers to effective public participation in the management of protected areas, Soc. Netw. Anal. Min., 8 (2018), p. 31, 10.1007/s13278-018-0509-y.

Response 4: The type of network analysis proposed in the article has been specified in the Introduction section.

Point 5: Minor comments:

Figure 21 is left in red and in polish I suppose. Is it the same figure with figure 20?

Response 5: Yes, this is the same figure. Figure 21 should not be included in the manuscript. We apologize for this editing error.

Point 6: Please keep on a single type of reference of figure in the ms – either fig. or figure.

Response 6: In the main text, figures are referred to as Figure 1, Figure 2, etc. The word “Figure” is abbreviated only in parentheses (e.g. Fig. 1), consistently throughout the manuscript.

Reviewer 3 Report

I accept the article in its current form

Author Response

RESPONSE TO REVIEWER 3

Point 1: I accept the article in its current form

Response 1: Thank you for accepting our manuscript.

This manuscript is a resubmission of an earlier submission. The following is a list of the peer review reports and author responses from that submission.

Round 1

Reviewer 1 Report

Thank you for the opportunity to review this article. The topic of the article is interesting and useful for GIS technicians involved in transport planning but in my opinion the paper itself has a low quality and a number of improvements must be done before publication of the article.

In addition, the problem that authors try to solve with their algorithm can be solved easily if the linear barriers are digitalized properly. Network analyst allows including linear barriers easily (as restriction or cost added) but not permeable zones. But If the barrier is split where a tunnel, bridge, the problem is solved. Then, a strong justification of the benefits of the presented tool must be clear.

Main comments:

Author says that “The proposed methodology had been described and tested in a previous study” is it not original? Then, there is a conflict here and it cannot be published again. But later in the text, the reader can understand that a previous algorithm has been modified. This must be clear in the text (now it is confusing). In addition, in this case, the description of the previous algorithm is so long. I recommend a revision of this methods section.

The introduction section must be extended identifying previous works using these service areas in order to find the gap that authors want to solve that justify the tool presented.

The results section is poor. The analysis of results must be extended describing properly the maps and the consequences of not using the tool. In addition, 4 cities are selected as case study. Why? This must be justified. The results and differences must be compared/related/analyzed according to the city typology.

Line 191. The use of the area is not correct. The borders of the polygons only link the points calculated by network analyst for an easily interpretation but they are not certain data. In fact, they can be easily modified in the network calculation properties.

The discussion section must include the contribution, identify the benefits and weakness and they must be compared with previous works.

In addition, a strong conclusions supported by results and discussion is needed.

Why the MAUP is an important issue? This must be extended.

How have the subareas been generated? (line 78)

Minor comments:

Abstract: change “also known as road access areas” by “also known as access areas”.

References number. the first one must be 1 (see line 15).

Line 23. Also pipelines, electric lines…

How the base points are selected?

Figure 5. Adjust scale bar to 5, 20 (ok), 10 and 10 km respectively.

Figure 7. Some text in Polish is included. Please, translate it.

 Figure 10. Use values as 10, 15,… in the scale bar.

Line 161. There is a typo mistake. The figures are 8-11.

The algorithm, script or model builder tool could be provided.

Reviewer 2 Report

The ms intitled “A methodology for generating service areas that accounts for linear barriers” is an interesting paper presenting the modification of an algorithm for mapping service areas. The manuscript is overall well written.  I made a few minor comments

-          Some figures are cited in the text after – i.e. figure 7-10, and some figures are not even mentioned in the text (figure 11) -  which makes it a little difficult to follow.

-          The MS should also contain a justification for choosing these particular case studies.

-          The method section should contain more details on the network analyses performed.

Reviewer 3 Report

This paper is well-written. Some comments need to be addressed for this paper to be published as follow:

You need to discuss why you specifically chose those base points. Do those points represent diverse cases?

What do you mean by preliminary analysis was done on Olsztyn, although the final results contain all of the 4 cities?
For example, for figure 6, Why did you illustrate Olsztyn only? You need to illustrate the rest of the cities.

I believe you need to rewrite the algorithm paragraphs in a more organized shorter paragraphs to make it easier for the reader.

For the results of the 4 cities (Figs 13-16), it would be beneficial if you could provide an aggregated single value that illustrates the deviation of service areas for your algorithm from ESRI algorithm for each city.

For the special cases in Figs 17 and 18, I believe you need to compare to ESRI algorithm to show the efficiency of your algorithm.

The conclusions part need to first summarize sufficiently the contribution of the paper before discussing the further research recommendations.

Reviewer 4 Report

The paper tackles a very interesting and important - for the transport accessibility studies – topic. This is another example that better data and tools giving possibility to measure accessibility more precisely.  Although the topic under analysis is an interesting and relevant one, there are some important shortcomings in the manuscript that prevent me recommend its publication in this journal.

My problem with the paper lies in the method of research. Author(s) mentioned in the text that method has been already published in the journal titled Roczniki Geomatyki (2017). After reading that paper I consider that there are some similarities between this two papers. I think author (s) should better prove the originality of the paper.

Other comments below:

1.     insufficient review of literature and methods and tools which measure transport accessibility.

2.     the results are not convincing, the author(s) do not sufficiently prove the superiority of their method over others methods

3.     Lack of discussion with the study results

Round 2

Reviewer 1 Report

Dear Authors,

Thanks for responding to the comments. I just suggest including a web link with the modelbuilder tool in order to be usefull for GIS technicians
  Best

Reviewer 4 Report

The author's comments and responses do not satisfy me. In my opinion explanation of differences between methodology in this paper and in the older one is still not enough. I also think that a literature review (authors added 3 papers?) as well as a description of the research results and conclusions are still insufficient.